# A tool for measuring mental workload during prosthesis use: The Prosthesis Task Load Index (PROS-TLX)

Johnny V. V. Parr[1], Adam Galpin[2], Liis Uiga[1], Ben Marshall[1], David J. Wright[3], Zoe C. Franklin[1], Greg Wood[1]*

1 Department of Sport and Exercise Sciences, Manchester Metropolitan University, Manchester, United Kingdom, 2 School of Health and Society, University of Salford, Manchester, United Kingdom, 3 Department of Psychology, Manchester Metropolitan University, Manchester, United Kingdom

* greg.wood@mmu.ac.uk

**Data Availability Statement:** All data relevant to this study are available from OSF at https://osf.io/

## Abstract

When using a upper-limb prosthesis, mental, emotional, and physical effort is often experienced. These have been linked to high rates of device dissatisfaction and rejection. Therefore, understanding and quantifying the complex nature of workload experienced when using, or learning to use, a upper-limb prosthesis has practical and clinical importance for researchers and applied professionals. The aim of this paper was to design and validate a self-report measure of mental workload specific to prosthesis use (The Prosthesis Task Load Index; PROS-TLX) that encapsulates the array of mental, physical, and emotional demands often experienced by users of these devices. We first surveyed upper-limb prosthetic limb users who confirmed the importance of eight workload constructs taken from published literature and previous workload measures. These constructs were mental demands, physical demands, visual demands, conscious processing, frustration, situational stress, time pressure and device uncertainty. To validate the importance of these constructs during initial prosthesis learning, we then asked able-bodied participants to complete a coin-placement task using their anatomical hand and then using a myoelectric prosthesis simulator under low and high mental workload. As expected, using a prosthetic hand resulted in slower movements, more errors, and a greater tendency to visually fixate the hand (indexed using eye-tracking equipment). These changes in performance were accompanied by significant increases in PROS-TLX workload subscales. The scale was also found to have good convergent and divergent validity. Further work is required to validate whether the PROS-TLX can provide meaningful clinical insights to the workload experienced by clinical users of prosthetic devices.

## Introduction

Using an upper-limb prosthesis limb is difficult. The absence of somatosensory feedback, impairments to motor control and problems with prosthesis fit can elevate levels of mental, physical (e.g., fatigue), and emotional (e.g., stress, frustration, and anxiety) effort, which have

s79gy/?view_only=
68b065d8c6ab4eecab8bbdc9d9b29325.

**Funding:** The author(s) received no specific funding for this work.

**Competing interests:** The authors have declared that no competing interests exist.

been linked to increased device dissatisfaction and high device rejection rates [1]. Several studies have suggested that this increased workload may be related to increased mental and physical demands [2], increased conscious attention to movement control [3, 4], high levels of frustration [5], stress [2] and high demands on visual attention [3, 4, 6, 7]. It is therefore clear that the workload experienced by prosthesis users is not a unidimensional construct, but a multifaceted, and probably individualistic, experience that spans across cognitive, physical, and emotional domains.

The importance of understanding the workload experienced during prosthesis use, and the methods available to measure it, has been underlined in recent reviews [1, 8]. Methods used to measure mental workload can be broadly split into physiological, performance and subjective domains. In the physiological domain measures such as electroencephalography (EEG) [3, 9], eye-tracking [3, 6], respiratory rate [10], skin conductance [10, 11] and cardiac markers, such as heart rate variability [10, 11], have all been used to infer the magnitude of workload. While physiological measures have the advantage of being objective, they are unidimensional in nature and lack detail in identifying the precise nature of the workload experienced. For example, increased brain activity, heart rate or pupil size could be related to increased physical effort (fatigue), increased stress, and/or increased conscious processing. Understanding the origin of increased workload is important and physiological measures cannot provide such information. Additionally, these methods are not suitable or practical for applied use in most clinical settings due the expense of acquiring them and the time and expertise required to use them efficiently and effectively.

In the performance domain, primary measures of task performance such as the speed of task completion, number of errors and reaction or response time can be used to infer the amount of effort exerted. Furthermore, measures of secondary task performances have been used to assess the remaining cognitive capacity of a prosthesis user when using their device [9, 11, 12]. In this paradigm, the successful performance of a concurrent secondary task is used to illustrate that the primary task of using the prosthesis does not exhaust attentional capacity. Although informative, like physiological measures, performance measures do not help to uncover the origin of the workload and do not reflect the multidimensional nature of workload experienced by prosthesis users. For example, just because a prosthesis user performs the task without errors or in a timely manner does not mean that the workload supporting such performance was low or that attentional demands were not high.

In terms of subjective measures of workload, the most widely used inventory of this nature is the National Aeronautics and Space Administration-Task Load Index [NASA-TLX; [13]]. The NASA-TLX was designed to discriminate between different types of workload by assessing the effort experienced across multiple dimensions, rather than viewing workload as a unidimensional construct. These dimensions are mental demand, physical demand, temporal demand, performance, effort, and frustration. The NASA-TLX is the most widely used tool in studies measuring workload during prosthesis use [8]. However, as it was originally developed for use with pilots during space flight, and has never been validated for prosthesis use, it is unlikely to reflect the unique demands experienced by prosthesis users. For example, previous research has suggested that the workload experienced during prosthesis use can be reflected in the adoption of conscious movement control strategies [3], high levels of stress [2] and a propensity to use vision to monitor the prosthesis while attempting to control it [4]. These indices of workload have been linked to device dissatisfaction and rejection but are not accounted for in the NASA-TLX.

In this paper we aimed to develop and validate a self-report measure specific to the multidimensional nature of the workload experienced during upper-limb prosthesis use—The Prosthesis Task Load Index (PROS-TLX). By adopting methodologies from previous research that

have designed and validated task-specific workload measures for surgery (SURG-TLX) [14], driving [15], and simulated (virtual) environments (SIM-TLX) [16], we first identified constructs of workload from current literature and upper-limb prosthesis user feedback. From this, we targeted those constructs related to the proficiency of prosthesis control which will be more meaningful to rehabilitation and technological advancements. Next, we sought to validate this inventory experimentally by manipulating sources of potential workload during upper-limb prosthesis use in abled-bodied participants using a prosthetic hand simulator. This enabled us to examine how experimental conditions differing in workload demands were reflected in changes in the constructs of the PROS-TLX.

## Method

### Designing the PROS-TLX

As the NASA-TLX is a well-validated instrument, the intention was to maintain its general structure but make it more relevant to the specific demands of prosthesis use. To do this we replicated the procedures used by Wilson et al. [14] and Harris et al. [16]. First, we consulted the literature that has documented some of the issues reported by prosthesis users [2–7, 12, 17–20] and collated these with existing dimensions of the NASA-TLX to best approximate the demands faced by device users. We then designed an online survey that eight (six male and two female) upper-limb prosthesis users completed to confirm that the workload dimensions proposed were reflective of the typical workload demands they faced and to confirm that these demands were reduced in relation to increased prosthesis control. This sample size is identical to previous research [14]. The mean age of the participants was 47.1yrs (SD = 8.29), seven of whom used a myoelectric hand prosthesis and one who used a body-powered hand prosthesis. Respondents had a mean of 15.3 years (SD = 9.53) experience of wearing their prosthesis. Three users wore their hands at least a few times each week and five reported using their prosthesis daily. The structure of the survey and anonymised responses are freely available for download from the Open Science Framework (OSF) (https://osf.io/s79gy/?view_only=68b065d8c6ab4eecab8bbdc9d9b29325). Participants were asked the following questions related to each construct and were asked to respond using one of five options: 'strongly disagree', 'disagree', 'neither agree or disagree', 'agree', or "strongly agree':-

1. Mental Demands: *To what extent do you agree that high levels of concentration and mental fatigue are typically experienced by patients learning to use a prosthetic hand*?

2. Physical Demands: *To what extent do you agree that physical fatigue (aching arms/muscles/soreness) is typically experienced by patients learning to use a prosthetic hand*?

3. Visual Demands: *To what extent do you agree that it is typical to feel the need to watch your prosthetic hand as you are learning to move it*?

4. Conscious Attention: *To what extent do you agree that you need to think consciously about how you are moving and controlling your prosthetic hand when first learning to use it*?

5. Frustration: *To what extent do you agree that high levels of frustration are typical when you are first learning to use a hand prosthesis*?

6. Situational Stress: *To what extent do you agree that stress and anxiety are typically experienced when first learning to use a hand prosthesis*?

7. Social Pressure: *To what extent do you agree that feeling pressure when using a prosthetic hand in front of people is typical when you are first learning to use a hand prosthesis*?

8. Time Pressure: *To what extent do you agree that feeling pressure to perform in a timely manner is typical when you are first learning to use a hand prosthesis*?

In addition to these questions, participants were asked the extent to which each construct reduced as they became more proficient at using their prosthesis. They chose from six response options from 'Not at all', 'A little', 'A moderate amount', 'A lot' 'A great deal' or 'I don't know'. This question was asked as we were interested in the indices of workload that were specifically related to increased user proficiency and which are likely to reduce during training and rehabilitation [8]. Free text comments were also allowed after each question and respondents were free to expand on any answer given or offer any other constructs that they deemed important.

Responses suggested that prosthesis users 'agreed' or 'strongly agreed' that Mental Demands (87.5%), Physical Demands (100%), Visual Demands (100%), Conscious Attention (87.5%), Frustration (100%), Situational Stress (50%), Social Pressure (87.5%), and Time Pressure (75%), were important constructs experienced when learning to use their prosthesis. Responses suggested that prosthesis users reported that Mental Demands (100%), Physical Demands (100%), Visual Demands (87.5%), Conscious Attention (87.5%), Frustration (87.5%), Situational Stress (50%), Social Pressure (50%), and Time Pressure (50%), all decreased by at least a 'moderate amount' with increased prosthesis control.

On exploration of the free text comments, users suggested that time pressure and situational stress were very task-specific. We decided to keep these constructs in the final version of the PROS-TLX as many tasks in research and rehabilitation will require timed performances and tasks that are pressurised (e.g., pouring water from a jar). The social stress construct was omitted from the final version as the free text comments revealed that this construct had little relevance to user proficiency and was more related to stress about the aesthetics of the device and others' knowledge about the personal circumstances around their disability. Although these are obviously important psychosocial aspects of a user's experience, they were not deemed to be relevant to the context for which the PROS-TLX is primarily designed (i.e., research related to workload and prosthesis control). Finally, respondents highlighted that trust or uncertainty in the reliability and consistency of their device's response was an issue when learning to use it; something which is also reported in recent research [19–21]. We therefore added an uncertainty construct to the inventory. The finalised PROS-TLX consisted of the following constructs: -

1. Mental Demands—How mentally fatiguing was using your prosthesis during that task?

2. Physical Demands—How physically fatiguing was using your prosthesis during that task?

3. Visual Attention—How much did you have to watch your prosthesis as you were moving during that task?

4. Conscious Processing—How much did you have to think about how you were moving during that task?

5. Frustration—How insecure, discouraged, irritated, stressed, or annoyed were you during that task?

6. Situational Stress—How anxious or stressed did you feel during the task?

7. Time Pressure—How hurried or rushed did you feel during that task?

8. Uncertainty—How unpredictable was your prosthesis during that task?

Questions 1, 2, 5 and 7 were taken and adapted from the original NASA-TLX. Having finalised the constructs of the PROS-TLX we then designed an experimental protocol to validate it.

## Validation of the PROS-TLX

**Participants.** Twenty-eight right-handed participants volunteered to participate in the study. The sample consisted of 17 males and 11 females (Mean age = 24.89, SD = 5.09 years). All reported normal or corrected-to-normal vision and gave written informed consent prior to testing. An institutional ethics committee granted approval of the experimental procedures prior to testing commencing.

## Apparatus

**Prosthetic hand simulator.** The prosthesis used in this study was the Bebionic™ (Otto Bock HealthCare, Duderstadt, Germany) fully articulating myoelectric prosthetic hand (Fig 1). To fit able-bodied participants, the hand was attached to a carbon fibre trough in which participants' forearm and wrist were positioned and fastened with Velcro straps. The myoelectric hand is controlled by isometric muscular contractions detected by two electrodes placed on the extensor (extensor carpi radialis) and flexor (flexor carpi radialis) muscles of the forearm. Activation of the extensor muscle triggers the opening of the hand whereas activation of the flexor muscle triggers the closing of the hand. Each participant was asked to open and close the hand until they could operate it when prompted on 10 consecutive attempts. The hand was pre-programmed into the 'tripod' grip, as is recommended in the manual for the Southampton Hand Assessment Procedure (SHAP) [22] for the coin task.

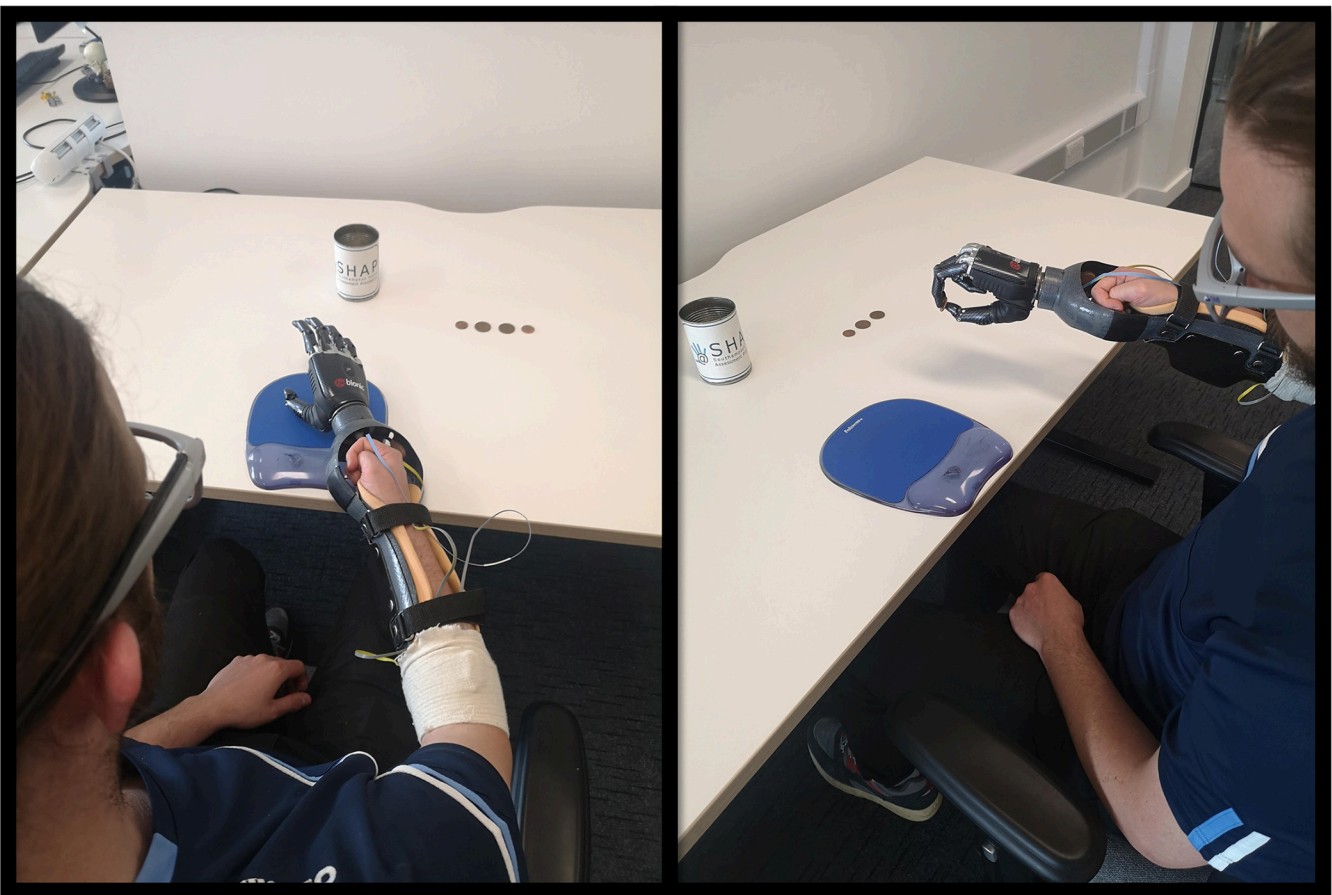

**Fig 1.** Showing the coin task setup (left) and the participant reaching to drop a coin into the jar (right).

**The coin task.** The SHAP is a clinically validated hand function test that was developed to assess the effectiveness of upper-limb prostheses [22]. For this experiment, the "picking up coins" task was used that from previous experience [3, 6] we know is very challenging and can often lead to frustration, places high demands on visual attention and is mentally, physically, and emotionally demanding. This sequential task required participants to pick up two UK 2 pence (2.6cm in diameter) and two UK 1 pence (2cm in diameter) coins by sliding them to the edge of the table before picking them up and placing them in a jar (Fig 1). If participants dropped a coin, it was replaced by the researcher.

**Eye-tracking.** Eye movements were monitored using ETG 2 w eye tracking glasses and iView ETG 2.7 software (SMI, Teltow, Germany). The system comprises a pair of lightweight glasses that track binocular eye movements at a sampling rate of 60 Hz with a gaze position accuracy of 0.5˚. The eye tracking glasses were calibrated for each participant prior to each condition by instructing them to fixate on points in the task environment (e.g., coins and jar).

**PROS-TLX.** The PROS-TLX kept the same two-part structure as the original NASA-TLX and other task-specific TLX measures that have been based upon the original [14, 16]. After completing each condition, participants rated their perception of workload on each of the eight dimensions (Part 1). These were reported on a 20-point Likert scale anchored from 1 (low) to 20 (high), like the original NASA-TLX. The relative importance (i.e., the weighting) of each dimension was captured by asking participants to make a series of 28 pairwise comparisons between dimensions (Part 2). For this, each dimension was paired against each other, and participants were asked to choose the construct that provided the most significant source of workload across each pair. To calculate a workload score for each dimension, the Likert scale score was multiplied by the weighting score. For example, a Likert rating score of 10 and a weighting score of 4 would achieve an overall dimension score of 40. A total workload score was calculated by totalling the workload score from each dimension and dividing it by the 8 constructs. For further clarity, an example of a completed inventory can be downloaded from OSF (https://osf.io/s79gy/?view_only=68b065d8c6ab4eecab8bbdc9d9b29325).

**Convergent and divergent validity.** To establish convergent validity, we took measures of overall effort using the Rating Scale of Mental Effort (RSME) [23]. This scale is a 0 to 150-point visual analogue scale containing anchor points with descriptive labels ranging from "absolutely no effort" through "considerable effort" to "extreme effort". As convergent validity relates to the strength of the relationship between two methods that are intended to measure the same underlying construct, we expected a strong positive correlation between RSME scores and those reported on the PROS-TLX.

Divergent validity tests whether concepts or measurements that are not supposed to be related are actually unrelated. To establish the divergent validity, we recorded measures of enjoyment using the enjoyment subscale of the Intrinsic Motivation Inventory (IMI) [24]. This scale consists of seven questions (e.g., "I enjoyed doing this activity very much") and responses were recorded on a 7-point Likert scale from 1 (not at all) to 7 (very true). The mean of all seven enjoyment scores was taken as an overall enjoyment score. We expected no significant correlation between enjoyment scores and PROS-TLX scores.

## Procedure

Participants attended the laboratory individually and after giving written consent the eye-tracker was calibrated to them. They were then given a demonstration of the coin task before completing 5 trials of 4 coins with their anatomical hand. They then completed both parts of PROS-TLX, the RSME and the IMI. Participants were then fitted with the prosthetic hand simulator and given instruction on how to operate it until they could open and close the hand on

demand. They were then allowed two practice attempts with 2 coins. They completed the prosthetic condition and repeated the coin task (i.e., 5 trials of 4 coins) with the prosthetic hand before completing Part 1 of the PROS-TLX, the RSME and the IMI. For both anatomic and prosthetic conditions participants were asked to move at a natural pace and focus on trying not to drop the coins rather than attempting to complete the task quickly. Finally, for the prosthesis + pressure condition, participants were told that they had two attempts to put all four coins into the jar as quickly as possible. They were told that this was a competition, that all scores would be tabulated and distributed between participants, and that the fastest individual would be given a £20 gift voucher. The primary aim of these task manipulations was to explore if the PROS-TLX subscales were sensitive to changes in task demands and to examine how these changes related to more objective measures like gaze behaviour. Participants then completed Part 1 of the PROS-TLX, the RSME and the IMI. Part 2 of the PROS-TLX was completed at the end of testing and participants were asked to complete the comparisons in relation to the workload experience from using the prosthesis in general rather than any one condition [16].

## Measures

**Performance.**   The mean time taken (secs) to complete each trial of 4 coins and the number of coin drops per trial were calculated across experimental conditions.

**Gaze behaviour.**   The number of fixations on the anatomical hand or prosthesis was calculated via frame-by-frame analysis using BeGaze analysis software (SMI, Teltow, Germany). A fixation was defined automatically by the software as any static gaze fixation over 80ms in duration. Each fixation was categorised as 'hand-focused' if it was located on the hand or on the coin as the hand was transporting it (as Parr et al., 2018; 2019). The number of hand fixations in a single trial were then divided by the total trial length, providing an index of mean hand fixation rate (per second) for each trial of 4 coins.

**Data analysis.**   As data were non-parametric, Friedman tests were conducted to explore the differences in each construct across Anatomic, Prosthesis and Prosthesis + pressure conditions. Bonferroni corrected Conover comparisons (test of multiple comparisons using rank sums as post hoc test following a significant Friedman test) were conducted to examine the differences between conditions. Spearman Rho correlation analyses were used to examine the overall relationship between all PROS-TLX constructs and between these constructs and measures of performance (i.e., mean task completion time and the number of coin drops) and eye-movements (mean number of hand-focused fixations). Further Spearman Rho correlation analyses were used to examine the convergent validity between overall PROS-TLX and RSME scores and the divergent validity between overall PROS-TLX and IMI scores. All analyses were conducted in JASP version 0.16.01 (JASP Team, 2021). All raw data and analysis outputs are available on the OSF (https://osf.io/s79gy/?view_only=68b065d8c6ab4eecab8bbdc9d9b29325).

## Results

### Performance and gaze behaviour

All gaze and performance data are presented in Fig 2.

**Performance time.**   A significant effect for condition, $\chi^2 (2) = 50.30$, $p < .001$, showed that performance time was significantly quicker in the anatomic condition compared to the prosthesis ($p < .001$) and prosthesis + pressure ($p < .001$) conditions. Performance time was also significantly quicker in the prosthesis + pressure condition compared to the prosthesis condition ($p = .009$).

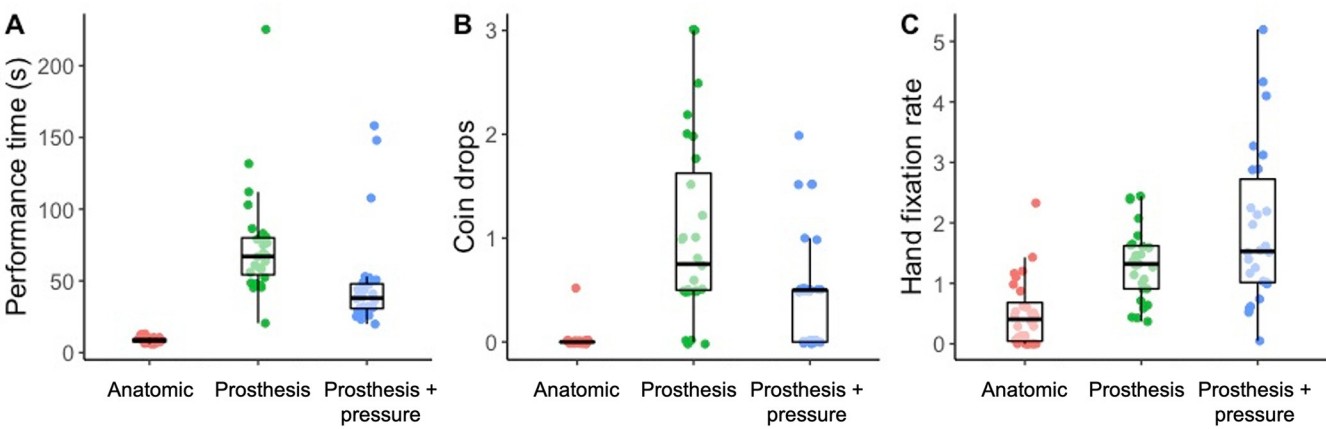

**Fig 2.** Boxplots displaying the median, quartiles, and each individual's mean performance time (A), number of drops (B) and hand fixation rates (C) across experimental conditions.

**Coin drops.** A significant effect for condition, $\chi^2$ (2) = 28.78, $p < .001$, showed that significantly fewer coin drops occurred in the anatomic condition compared to the prosthesis ($p < .001$) and prosthesis + pressure ($p = .005$) conditions. The number of coin drops was not significantly different between prosthesis and prosthesis + pressure conditions ($p = .160$).

**Hand fixation rate.** A significant effect for condition, $\chi^2$ (2) = 19.00, $p < .001$, showed that the number of hand-focused fixations were significantly less in the anatomic condition compared to the prosthesis ($p = .023$) and prosthesis + pressure ($p < .001$) conditions. No significant difference was found between prosthesis and prosthesis + pressure conditions ($p = .400$).

## PROS-TLX

All PROS-TLX data are presented in Fig 3.

**Mental demands.** A significant effect for condition, $\chi^2$ (2) = 38.74, $p < .001$, showed that the mental demands were significantly lower in the anatomic condition compared to the prosthesis ($p < .001$) and prosthesis + pressure ($p < .001$) conditions. No significant difference was found between prosthesis and prosthesis + pressure conditions ($p = .842$).

**Physical demands.** A significant effect for condition, $\chi^2$ (2) = 43.85, $p < .001$, showed that the physical demands were significantly lower in the anatomic condition compared to the prosthesis ($p < .001$) and prosthesis + pressure ($p < .001$) conditions. No significant difference was found between prosthesis and prosthesis + pressure conditions ($p = 1.00$).

**Visual demands.** A significant effect for condition, $\chi^2$ (2) = 23.94, $p < .001$, showed that the visual demands were significantly lower in the anatomic condition compared to the prosthesis ($p = .003$) and prosthesis + pressure ($p < .001$) conditions. No significant difference was found between prosthesis and prosthesis + pressure conditions ($p = .670$).

**Conscious processing.** A significant effect for condition, $\chi^2$ (2) = 32.80, $p < .001$, showed that conscious processing was significantly lower in the anatomic condition compared to the prosthesis ($p < .001$) and prosthesis + pressure ($p < .001$) conditions. No significant difference was found between prosthesis and prosthesis + pressure conditions ($p = 1.00$).

**Frustration.** A significant effect for condition, $\chi^2$ (2) = 30.43, $p < .001$, showed that frustration was significantly lower in the anatomic condition compared to the prosthesis ($p < .001$) and prosthesis + pressure ($p < .001$) conditions. No significant difference was found between prosthesis and prosthesis + pressure conditions ($p = .499$).

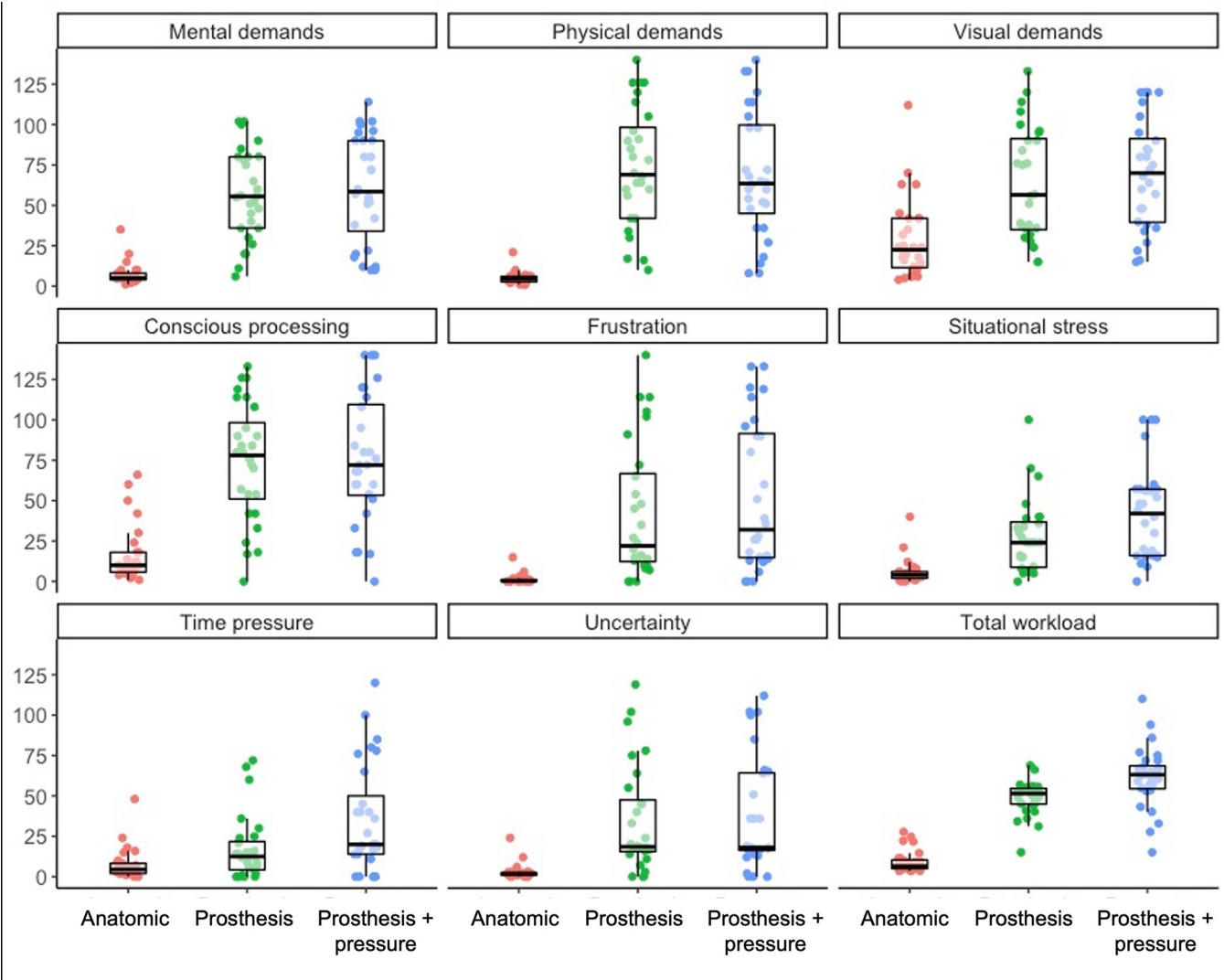

**Fig 3. Boxplots displaying the median, quartiles, and each individual's mean for PROS-TLX subscales and total workload scores across experimental conditions.**

**Situational stress.**   A significant effect for condition, $\chi^2$ (2) = 35.39, $p < .001$, showed that situational stress was significantly higher in the prosthesis + pressure condition compared to the anatomic ($p < .001$) and prosthesis conditions ($p = .002$). There was no significant difference in the situational stress experienced between the anatomic and prosthesis conditions ($p = .088$).

**Time pressure.**   A significant effect for condition, $\chi^2$ (2) = 23.79, $p < .001$, showed that time pressure was significantly higher in the prosthesis + pressure condition compared to the anatomic ($p < .001$) and prosthesis condition ($p < .001$). However, time pressure was not significantly different in the anatomic condition compared to the prosthesis ($p = 1.00$) conditions.

**Uncertainty.**   A significant effect for condition, $\chi^2$ (2) = 24.52, $p < .001$, showed that uncertainty was significantly lower in the anatomic condition compared to the prosthesis ($p < .001$) and prosthesis + pressure ($p < .001$) conditions. No significant difference was found in uncertainty between prosthesis and prosthesis + pressure conditions ($p = 1.00$).

**Total workload.** A significant effect for condition, $\chi^2$ (2) = 43.05, $p < .001$, showed that the total effort was significantly lower in the anatomic condition compared to the prosthesis ($p < .001$) and prosthesis + pressure ($p < .001$) conditions. Total effort was also significantly lower in the prosthesis condition compared to the prosthesis + pressure condition ($p = 0.041$).

## Correlations

Analysis of the relationships between PROS-TLX constructs showed moderate to strong correlations between almost all PROS-TLX dimensions and every dimension correlated with total workload score, as expected (see Fig 4). Similar significant relationships between performance and workload scores were also evident (Fig 5).

## Convergent and divergent validity

A significant positive correlation was found between total workload and RSME scores, $r_s = .87$, $p < .001$. This suggests the PROS-TLX has good convergent validity with another established measure of mental effort. No significant correlation was found between total workload and IMI scores, $r_s = .10$, $p = .391$. This suggests the PROS-TLX has good divergent validity with an unrelated measure of enjoyment. These data support the construct validity of the PROS-TLX (Fig 6).

## Discussion

The aim of this paper was to design and validate a self-report measure of mental workload specific to upper-limb prosthesis use that encapsulates the cognitive, physical and emotional

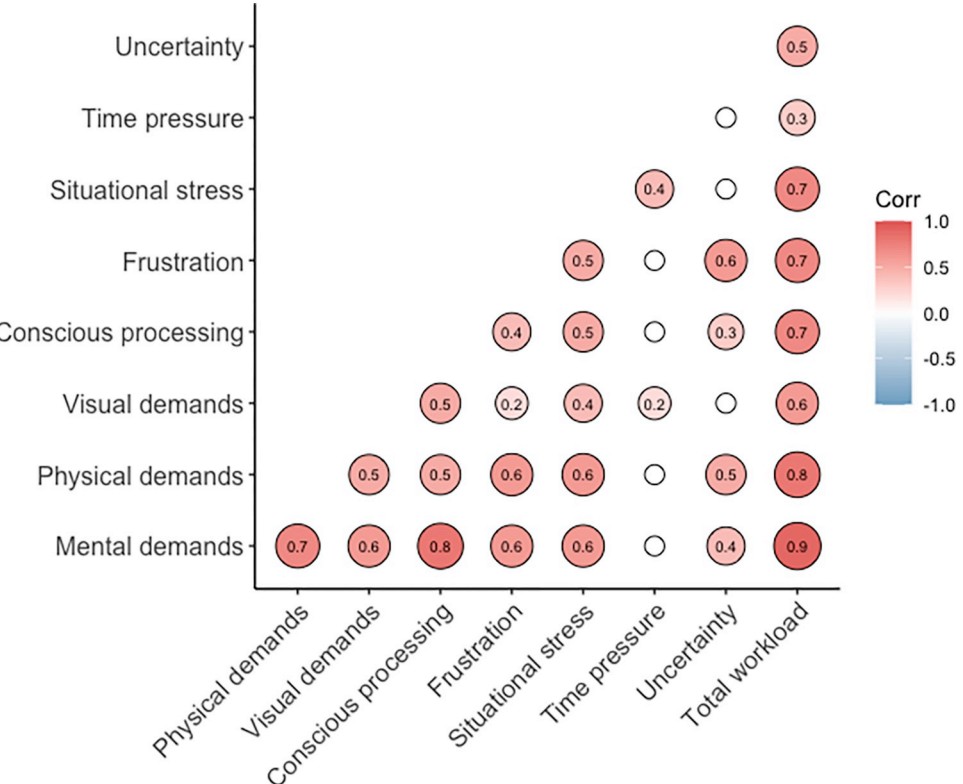

**Fig 4. Correlogram showing Spearman Rho correlations between PROS-TLX constructs and total workload score.** Values written within each circle represent a significant $r^2$ value at the $p < .05$ level. Smaller white circles without a numerical value represent non-significant correlations.

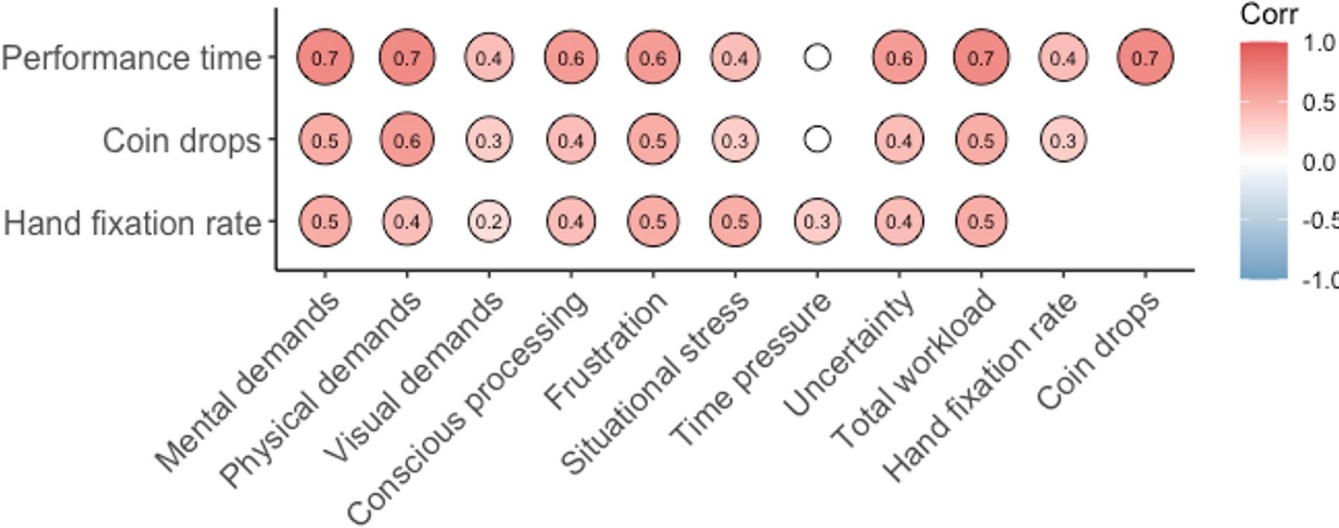

**Fig 5. Correlogram showing Spearman Rho correlations between performance time, coin drops, gaze behaviour with each PROS-TLX construct and total workload.** Values written within each circle represent a significant $r^2$ value at the $p < .05$ level. Smaller white circles without a numerical value represent non-significant correlations.

demands often experienced by users of these devices. Results showed that the experimental manipulations elicited significant changes in behaviour across the experimental conditions. Performance times were slower, more coins were dropped and there were more hand-focused fixations when participants used the prosthesis compared to when they used their anatomical hand (Fig 2). In the prosthesis + pressure condition, participants dropped fewer coins and performed quicker compared to the prosthesis condition. However, the rate at which they fixated the hand remained unchanged. These changes in behaviour also translated to significant changes to workload.

There was a significant increase in workload scores for every scale of the PROS-TLX, except time pressure, when participants used the prosthesis compared to when they used their

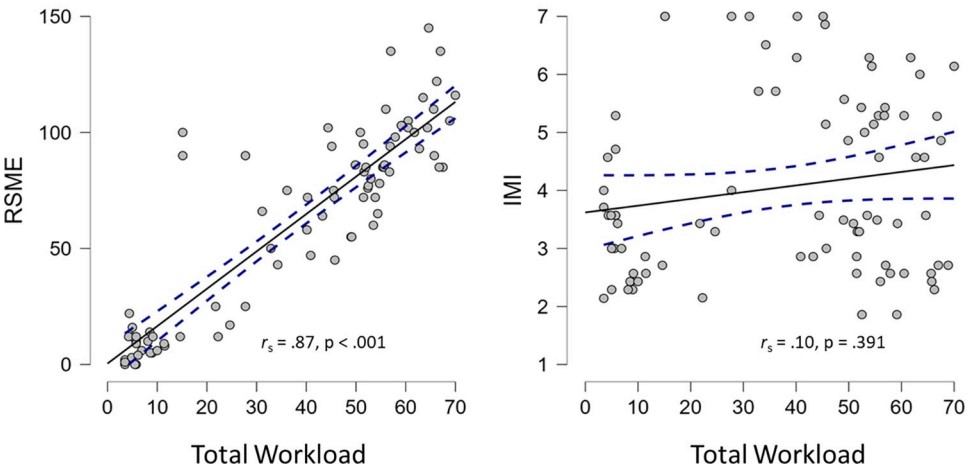

**Fig 6.** Scatter plots showing the convergent (left) and divergent (right) relationships between the total workload score and the RSME and IMI. Dotted lines represent the 95% CI.

anatomic hand, as expected (Fig 3). As participants were asked to perform in a natural, controlled manner in both the anatomic and prosthesis conditions no differences in the perceptions of time pressure were expected. The prosthesis + pressure condition significantly increased workload scores in relation to situational stress and time pressure compared to both anatomic and prosthesis conditions. These increases are what contributed to the significant increase in total workload scores, and we suggest this may have heightened motivation and effort to perform well, resulting in quicker performance times in this prosthesis + pressure condition [25]. Together these results suggest that the PROS-TLX constructs encapsulate the workload characteristics experienced when using a prosthesis simulator and suggest that the PROS-TLX is sensitive enough to measure them.

Some notable relationships, that are consistent with findings from empirical research, were also evident (see Figs 4 and 5). For example, the relationship between visual demands and conscious processing are consistent with patient reports [18] and experimental research which has shown that when users watch their prosthesis, they are often investing cognitive effort [3]. It has been proposed that such a reliance on visual attention is due to the user trying to formulate sensorimotor mapping rules that govern prosthesis control [6] and/or due to the uncertainty inherent within the device itself [21]. Visual demands were also correlated to time pressure suggesting that the more pressure a user experiences the more they are likely to watch their hand. Theoretical justification for the direction of this relationship can be gained from research that shows increased pressure, and the resultant stress likely experienced, causes individuals to invest in behaviours designed to safeguard performance like increased visual attention to movement control or covert conscious movement control strategies [26]. The overall reliance on vision is also reflected in significant correlations with total workload scores, the number of drops and performance time. However, it is notable that the relationship between perceptions of visual demands and hand fixation rate was weak ($r = .02$). A potential reason for this may be due to the fact eye-trackers only capture the overt allocation of gaze and maybe participants were using covert attentional strategies (i.e., peripheral vision) to monitor hand location.

While there are too many significant relationships between PROS-TLX constructs to discuss individually, we will attempt to contextualise how these relationships may relate to increased workload faced by prosthesis users. For example, the responsiveness of a myoelectric hand prosthesis can be disrupted by poor socket fitting and increased socket sweating, leading to slower performance times, device uncertainty, and frustration. Device uncertainty causes frustration which then leads the user to attempt to consciously control and watch the prosthesis whilst trying to use it. Alternatively, the weight of the prosthesis may elevate physical fatigue, impairing myoelectric control and the responsiveness of the device. This could lead to increased stress and frustration, the exertion of conscious control processes, and ultimately slower and more error-strewn performance. These examples are supported by the correlations of PROS-TLX subscales and can explain how workload may be increased for a prosthesis user. While these are just two examples they illustrate why a prosthesis-specific measure of the multidimensional nature of workload is warranted.

Although the findings of this initial validation are encouraging there are several limitations and future research directions that need to be outlined. First, the survey respondents were primarily myoelectric upper-limb prosthesis users. It could therefore be argued that users of different devices (e.g., body-powered) or with differing levels of amputation may have different experiences with each workload construct. Second, although the central constructs of the PROS-TLX were based on recurring themes from across literature taken from both lower [2, 5, 9, 12, 19] and upper-limb prosthesis users [3, 4, 6, 7, 17, 18, 20], we do not know if the PRO-TLX could captured the nature of workload experienced in lower-limb prosthesis users. While the individual importance of each workload construct may differ across users of

different devices or across upper and lower-limb clinical patients, we believe that the importance of these constructs should still be relevant to prosthesis use in general, certainly more relevant than the most widely used measure of workload in prosthesis users, the NASA-TLX, which was designed for pilots during space flight. It is clear, however, that much more work is needed in validating this inventory across such populations.

Another important question that is unclear at this point is if these workload constructs reduce with increased device proficiency over time. While the survey respondents all suggested that they did, the risk of recall bias of respondents attempting to accurately remember changes in sources of workload across their 15 years of device use is a clear limitation of the study. Much more research is needed in attempting to understand the relationship between mental workload, user proficiency and measures of device satisfaction when using the device for the first time and also across longer periods of time. A tool like the PROS-TLX, that is quick to administer, may prove useful for this purpose.

Finally, we have only validated this workload measure in able-bodied participants using a simulator so it is possible that the results might not reflect the workload experienced in the clinical population. While this is a notable limitation of this work, previous research using prosthesis simulators has shown comparable kinematic profiles [27] visuomotor behaviours [3, 6] and perceptual experiences [28] to prosthesis users, suggesting that using a simulator provides a useful surrogate to provide an insight into the sensory-motor deficits that prosthesis users face when learning and using a prosthesis [29]. That said, further validation in clinical users is needed and future research should seek to apply the PROS-TLX during the rehabilitation of amputees and in long-term prosthesis users. Such work will highlight how the multidimensional nature of workload changes over time in response to training, cognitive and physical capacity, and technological advancement.

In conclusion, here we present the PROS-TLX which we have shown has convergent and divergent validity and captures the multidimensional nature of the cognitive, physical, and emotional workload experienced by able-bodied users of a myoelectric prosthesis simulator. Whilst the workload demands experienced by able-bodied users of a prosthesis simulator are unlikely to be identical to those experienced by real prosthesis users, they do provide insight into the general workload demands associated with learning to control a myoelectric device. The PROS-TLX, therefore, likely represents a more appropriate measure for assessing prosthesis workload than the NASA-TLX. As such, we believe that this tool will predominantly be used by researchers in understanding prosthesis control, testing interventions, or to supplement, and enrich, other physiological indices of workload. However, clinicians may also wish to use the PROS-TLX to monitor the nature of the workload experienced and as a starting point for further dialog around addressing the challenges patients face during rehabilitation. Finally, designers of prosthesis technologies may find it beneficial to use the tool to assess how developing technologies, such as the addition of prosthesis 'vision' [30] or augmented feedback systems (e.g., vibrotactile feedback), can impact the workload experienced by prosthesis users. Understanding the mental workload experienced by prosthesis users continues to be a major challenge for those working in this area, we hope this initial validation of the PROS-TLX is a major step forward in addressing this challenge. Moving forward, we invite researchers and applied practitioners to use the PROS-TLX with clinical patients and those using different prosthetic devices to strengthen the support for the use of this inventory.

## Acknowledgments

The authors would like to thank Meghann Leaver for her help with data collection and the analysis of the gaze data during this project.

## Author Contributions

**Conceptualization:** Johnny V. V. Parr, Adam Galpin, Liis Uiga, Greg Wood.

**Data curation:** Greg Wood.

**Formal analysis:** Johnny V. V. Parr, Greg Wood.

**Funding acquisition:** Liis Uiga, Ben Marshall.

**Investigation:** David J. Wright, Zoe C. Franklin, Greg Wood.

**Methodology:** Johnny V. V. Parr, Adam Galpin, Liis Uiga, Ben Marshall, David J. Wright, Zoe C. Franklin, Greg Wood.

**Project administration:** Greg Wood.

**Resources:** Greg Wood.

**Software:** Ben Marshall.

**Supervision:** Liis Uiga.

**Visualization:** Johnny V. V. Parr, David J. Wright.

**Writing – original draft:** Johnny V. V. Parr, Adam Galpin, Liis Uiga, Ben Marshall, David J. Wright, Zoe C. Franklin, Greg Wood.

**Writing – review & editing:** Adam Galpin, Liis Uiga, Ben Marshall, David J. Wright, Zoe C. Franklin, Greg Wood.

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
